# Fabrication of Silicon Nanowires by Metal-Assisted Chemical Etching Combined with Micro-Vibration

**DOI:** 10.3390/ma16155483

**Published:** 2023-08-05

**Authors:** Weiye Huang, Junyi Wu, Wenxin Li, Guojin Chen, Changyong Chu, Chao Li, Yucheng Zhu, Hui Yang, Yan Chao

**Affiliations:** 1School of Mechanical Engineering, Hangzhou Dianzi University, Hangzhou 310018, China; hzdzkjdx.hwy@hdu.edu.cn (W.H.); wwenxindiaolong@163.com (W.L.); chenguojin@163.com (G.C.); kevin@hdu.edu.cn (C.C.); zhuyucheng202205@163.com (Y.Z.); 222010073@hdu.edu.cn (H.Y.); 2Sanmen Sanyou Technology Inc., Taizhou 472000, China; willian199109@163.com

**Keywords:** micro-vibration platform, MaCE, etching rate, amplitude, frequency

## Abstract

In this work, we design a micro-vibration platform, which combined with the traditional metal-assisted chemical etching (MaCE) to etch silicon nanowires (SiNWs). The etching mechanism of SiNWs, including in the mass-transport (MT) and charge-transport (CT) processes, was explored through the characterization of SiNW’s length as a function of MaCE combined with micro-vibration conditions, such as vibration amplitude and frequency. The scanning electron microscope (SEM) experimental results indicated that the etching rate would be continuously improved with an increase in amplitude and reached its maximum at 4 μm. Further increasing amplitude reduced the etching rate and affected the morphology of the SiNWs. Adjusting the vibration frequency would result in a maximum etching rate at a frequency of 20 Hz, and increasing the frequency will not help to improve the etching effects.

## 1. Introduction

SiNWs are attracting increasing attention due to the performance of optoelectronic devices improved by integrating even the simplest silicon nanostructures, and they are documented as the building block in the fields of optoelectronics [1,2], microelectronics [3,4], solar cells [5,6,7], and chemical and biological sensors [8]. Many efforts have been made to develop methods to fabricate the SiNWs. Until now, dry etching methods and wet etching have been the major methods for SiNW fabrication. Dry etching, including reactive ion etching (RIE) [9,10], vapor–liquid–solid method (VLS) [11,12], and plasma ion-coupled reactive ion etching (PICRIE), can fabricate high-quality SiNWs. However, their vast application in the industry is hindered by the high cost of instrumentation. Wet chemical etching [13,14] can fabricate SiNWs with low-cost alternatives, but it is prone to isotropic etching, and the qualities of SiNWs can be influenced by the etching solution’s concentration and composition. MaCE plays an essential role in the preparation of SiNWs due to its simplicity, versatility, and cost-effectiveness compared with other etching methods. However, the etching environment change, including the etchant concentration and generation of etching production, will continuously influence the etching rate and the quality of SiNWs.

Metal-assisted chemical etching (MaCE) [15,16,17,18], developed by Li and Bohn in 2000 [19], is a powerful etching technique for fabricating SiNWs with simplicity, versatility, and cost-effectiveness compared to dry and wet etching methods. In the MaCE process, the thin layer or particle of noble metal (e.g., Ag, Au, etc.) acts as a catalyst in a solution mixed with hydrogen peroxide and hydrofluoric acid to etch the SiNWs [20]. Under the action of a catalyst, hydrogen peroxide is an oxidant and continuously oxidizes silicon, while the oxidized silicon is constantly absorbed by hydrofluoric acid. As a redox process, MaCE allows for the anisotropic etching of the silicon directly underneath the catalyst metal, leaving behind the patterns of noble metal on the silicon substitute.

During the MaCE etching, the required etchant, including oxidants and acids, is achieved by molecular diffusion, which is influenced by the morphology formed on the silicon, continuously decreasing the diffusion rate. At the same time, the decrease in etchant concentration affects the MaCE etching rate. Meanwhile, the products of the etching process, due to their high density, reside at the interface between the catalyst and silicon, which also affects the etching [21,22,23]. Li et al. proposed a method to improve the diffusion rate of the etchant and the etching environment by stirring the etchant solution. However, during the stirring process, noble metals, presented as meshes or particles, are also carried out or deviated from their original positions with the flow of the etchant, thereby affecting the quality of SiNWs.

In this work, we propose an innovative method by introducing micro-vibrations into MaCE to improve the diffusion rate of etchant and product in MaCE, as mentioned above. A micro-vibration platform with adjustable amplitude and frequency has been designed. On this platform, a silicon wafer with noble metal catalysts is vibrated slightly in the reaction solution instead of stirring the solution to avoid the issues in Li’s research.

The effect of micro-vibration on etching SiNWs was studied by changing the parameters of micro-vibration. Si samples etched at different parameters are characterized. We have found that micro-vibrations facilitate the transport of etchants and products. Moreover, the problem of the accumulation of products at the root of the nanowires was solved, and the rate of etching of nanowires was improved.

## 2. Micro-Vibration Platform

In the field of aerospace [24], micro-vibration caused by disturbance sources of spacecraft can seriously damage the working environment of sensitive loads. In recent years, a number of well-known vibration control methods have been developed, especially for suppressing or isolating micro-vibration, and passive isolation techniques are commonly used in aerospace engineering [25]. However, in this work, we use micro-vibration to conduct experiments and design a micro-vibration device based on the existing literature to investigate the effects of micro-vibration on the MaCE etching of SiNWs [26,27,28,29,30].

The micro-vibration platform needs to realize different forms of vibration, and the influence of MaCE on the rate and quality of etching SiNWs under different amplitudes and different frequencies of micro-vibration is studied. Therefore, it is necessary to choose a signal generator that can generate current signals with different functions, and the frequency and amplitude are adjustable. Because the power of the signal generator is small, the current generated after the output signal may be too small, resulting in the vibration rod cannot be started, so the power amplifier used in conjunction with this should be selected. The FY2300-06M signal generator is selected as the source signal. FPA1016 power amplifier is used to amplify the source signal. Detailed parameters of the function signal generator and power amplifier are shown in Table 1 and Table 2.

The structure of the micro-vibration platform is illustrated in Figure 1. The micro-vibration platform mainly consists of the signal generator, signal amplifier, vibration device, etc.

The vibration signal is yielded by the signal generator. Followed by an amplified signal amplifier, the signal is inputted into the coil of the vibration device to generate the induced magnetic field, which drives the armature installed inside the coil to vibrate. Since the rigid connection between the armature and the vibrating rod, the vibrating rod is vibrated together with the armature. When the silicon-containing noble metal in the reactor is connected to the vibrating rod, the combination of MaCE with micro-vibration is achieved. To ensure the accuracy of the vibration, sensors are installed near the vibration rod to detect vibration amplitude and frequency, and the detection results are fed back to the signal generator. Vibration signals with different amplitudes, frequencies, and vibration types are generated by changing all kinds of input parameters to meet the requirements of users.

The vibrating rod is made of engineering plastic and ABS resin, which can reduce the weight of the rod and improve the sensitivity of vibration. On the other hand, engineering plastics are a kind of magnetic insulating material which can effectively isolate the influence of magnetic fields and ensure the stability of vibration. In addition, two pairs of highly sensitive springs are arranged symmetrically with the horizontal surface at 5° to connect the vibrating rod to the housing. When the vibrating rod vibrates back and forth, one pair of springs generates tension, and the other pair of springs generate pressure, effectively reducing the influence of the inertia force generated by the weight of the vibrating rod. The spring can reduce the impact of the vibrating rod, ensure the reliability of the vibration, and improve the accuracy of the displacement.

## 3. Experimental

Polished single-sided crystalline (111) p-type boron-doped Si wafers with a resistivity of 0.004–0.007 Ωcm were cleaved into 1 × 1 cm^2^ samples. Silicon samples were washed in acetone and HF (4.8 mol/L) solution for 10 min to remove grease and SiO_2_ from the sample. Ag nanoparticles were deposited on Si samples by immersion in a mixture of AgNO_3_ (0.01 mol/L) and HF (4.8 mol/L) solution for 1 min.

After chemical metallization, the Si wafers were immersed in HF (4.8 mol/L) and H_2_O_2_ (0.2 mol/L) solution in a reactor on the micro-vibration platform. The device was started, the Si wafers vibrated back and forth under different forms of micro-vibration, and the SiNWs were etched. The parameters of each experimental group for micro-vibration combined with the MaCE etching of SiNWs are shown in Table 3.

After etching, the samples were rinsed with deionized water and dried. The images of samples were recorded by a scanning electron microscope (GeminiSEM 300) to study nanowire structures (The SEM is manufactured by Zeiss in Germany and purchased from Carl Zeiss Shanghai Management Co., Ltd., Shanghai, Chnia).

## 4. Results and Discussion

Figure 2 shows the SEM cross-sectional view of SiNWs. The SiNWs in Figure 2a were etched by MaCE without micro-vibration, while those in Figure 2b–d were etched by MaCE combined with micro-vibration with amplitudes of 2 μm, 4 μm and 6 μm, respectively. The micro-vibration frequency is fixed at 20 Hz, and the etching time of all samples is 30 min.

It could be found from Figure 2 that the length of SiNWs in Figure 2b–d is longer than the length of SiNWs in Figure 2a. This result indicates that the introduction of micro-vibration does not destroy the formation of SiNWs, and micro-vibration helps to improve the depth of MaCE etching SiNWs. According to the etching mechanism of MaCE, micro-vibration accelerates the etching rate of SiNWs, and the result could be analyzed as follows:

From the view of the mechanism [31], the process of MaCE can be divided into two parts: the mass-transport (MT) and charge-transport (CT) processes. MT includes the transport of reactants (HF and H_2_O_2_) and products (H_2_O and H_2_SiF_6_), while CT consists of the generation, transportation, and consumption of h^+^.

Cathode reaction (at metal):H_2_0_2_ + 2H^+^ → 2H_2_0 + 2h^+^(1)
2H^+^ + 2e^−^ → H_2_(2)

Anode reactions (underneath the Ag nanoparticles):Si + 2H_2_0 → SiO_2_ + 4H^+^ + 4e^−^(3)
SiO_2_ + 6HF → H_2_SiF_6_ + 2H_2_0(4)

The introduction of micro-vibration promotes the exchange between reactants and reaction products in the etching process (MT), and the etching model at this stage is shown in Figure 3b. Under micro-vibration, the hydrofluoric acid and hydrogen peroxide required for the reaction will flow along the side wall of the SiNWs into the bottom of the SiNWs, while the silicofluoric acid product will flow from the side wall of SiNWs to the top of silver nanoparticles with the vibration of silver nanoparticles. The silicofluoric acid product no longer accumulates under the silver nanoparticles and does not hinder the transport of the etching agent and affect the etching process. Therefore, the depth of SiNWs etched under micro-vibration is much higher than that obtained by conventional MaCE etching.

We measured the length of SiNWs in Figure 1 and plotted it as a line table, as shown in Figure 4, showing the etching rate of SiNWs with different amplitudes. The etching rate in Figure 4 shows a trend of first increasing and then decreasing. The etching rate reaches its maximum value at the amplitude of 4 μm. The reason is that the larger the amplitude, the more reactants obtained (MT and CT) and the higher the etching rate. The etching model at this stage is shown in Figure 3c. However, when the amplitude exceeds 4 μm, the amplitude is too large, resulting in silver nanoparticles easily being carried away from the bottom of the SiNWs and even reaching the silicon surface. At the same time, due to the laminar flow generated by vibration, silver nanoparticles will be pushed to new positions and form new SiNWs, which affects the etching rate of SiNWs, and the reaction model at this time is shown in Figure 3d. This can also be supported by the numerous thin and short nanowires shown in Figure 2d. Meanwhile, in Figure 2d, regular nanowires can be found in the middle of the silicon wafer, while the surrounding areas are disorderly. This is also related to the small effect of laminar flow on the middle of silicon, while the large effect is around it. As a result, the silver nanoparticles around the silicon wafer are constantly taken away from their original positions and etched into new SiNWs.

Figure 5 is a planned SEM image of the SiNW structure. Figure 5a shows the SiNWs prepared by the MaCE without vibration, while Figure 5b shows the SiNWs prepared by the MaCE combined with micro-vibration at the frequency of 20 Hz and amplitude of 4 μm. The etching time of all samples is 30 min.

By comparing Figure 5a and Figure 5b, it can be seen that the SiNW structure in Figure 5b is more evenly distributed and has no obvious defects. A possible reason is that in the absence of vibration, silver nanoparticles deposited on silicon form an unevenly distributed and stacked distribution state. However, in the process of micro-vibration combined with MaCE etching, only a small amount of silver nanoparticles remained on the surface of the silicon wafer as the silver nanoparticles acted as a catalyst. At the same time, under the action of continuous micro-vibration, these stacked silver nanoparticles can be split into small pieces of relatively uniform size so that the distribution of SiNWs generated by MaCE etching is relatively uniform and the size is consistent.

We also observed the cross-sectional view SEM image of SiNWs etched under micro-vibration at the frequency of 40 Hz and amplitude of 2 μm, as shown in Figure 6. It can be seen that the depth of SiNWs etched at the frequency of 40 Hz is lower than the depth of SiNWs etched at the frequency of 20 Hz, and the SiNWs are arranged obliquely (see Figure 2b). In order to more clearly reflect the etching state in these two cases, we present a model as shown in Figure 7.

The low-frequency micro-vibration makes the contact time between silver nanoparticles and silicon wafers sufficient, and the number of electron holes generated by the reaction is sufficient to supply the etching process, thus ensuring the etching rate, and the reaction model at this stage is shown in Figure 7a.

The reason for the analysis of the situation shown in Figure 6 is that when the frequency exceeds a certain range, although the high-frequency micro-vibration aggravates the transport process of reactants and reaction products (MT), it will reduce the contact time between silver nanoparticles and silicon wafers, resulting in a decrease in the number of electron holes generated by the reaction (CT), thus reducing the etching rate, and the reaction model at this stage is shown in Figure 7b (the thickness of the arrow represents the rate of CT transmission). Eventually, the etching rate of SiNWs is affected by high-frequency micro-vibration, and the etching depth of SiNWs is reduced. In addition, the higher frequency enhances the randomness of the motion of silver nanoparticles, which leads to the possibility of adsorption of silver nanoparticles on SiN sidewalls. As a result, the oblique SiNW structure is etched.

## 5. Conclusions

In this work, the method of micro-vibration combined with MaCE is studied to etch SiNWs. We designed a micro-vibration device to meet experimental requirements. Different micro-vibrations are controlled to study the effects of amplitude and frequency on SiNW manufacturing. The scanning electron microscopy (SEM) results show that the micro-vibration can improve the etching rate of SiNWs by MaCE. At the same time, in the range of 0~4 μm, the etching rate of SiNWs increased, and when the amplitude exceeded 4 μm, the etching rate began to decrease. SiNWs show good etching quality in the low-frequency vibration range, but in the high-frequency vibration range, the structure becomes irregular after SiNWs etching. Etching silicon nanowires with micro-vibration is a method that has yet to be proposed. Our research has a positive significance for the preparation of nanomaterials and provides a creative idea for the preparation of silicon nanowires with a high aspect ratio structure.

## Figures and Tables

**Figure 1 materials-16-05483-f001:**
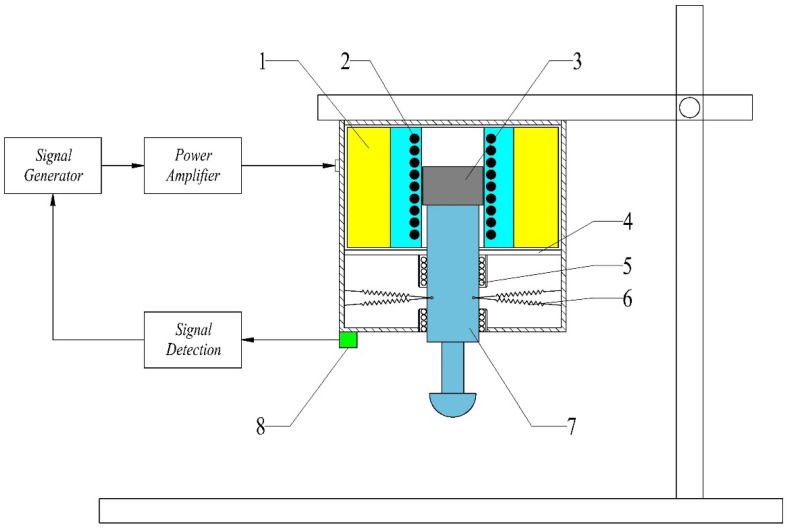
Equipment of micro-vibration platform; 1—Engine Bed; 2—Coil; 3—Armature; 4—Aluminum Plate; 5—Ball Bearing; 6—Spring; 7—Vibrating Rod; 8—Sensor.

**Figure 2 materials-16-05483-f002:**
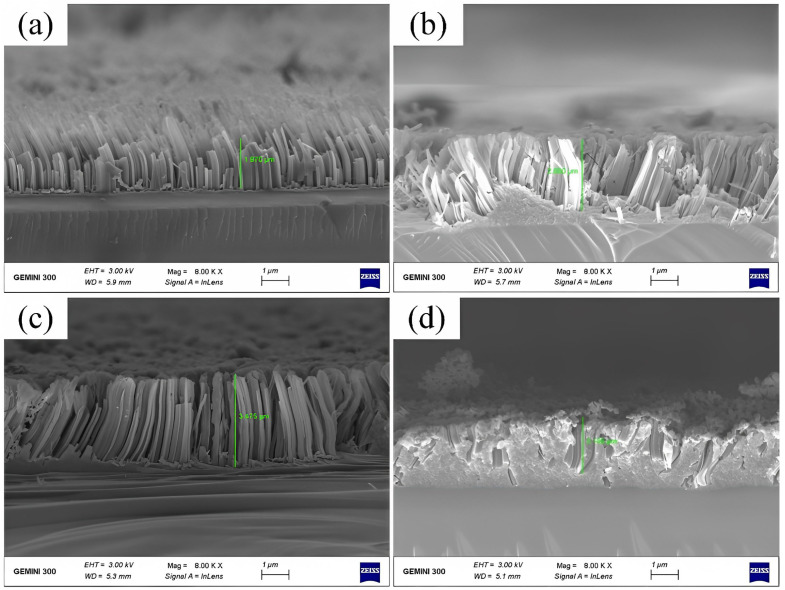
Cross-sectional view SEM image of SiNWs with no vibration (**a**), and with the vibration at the amplitude of 2 μm (**b**), 4 μm (**c**), and 6 μm (**d**). The numerical values marked in green are 1.870 μm, 2.680 μm, 3.475 μm and 2.135 μm.

**Figure 3 materials-16-05483-f003:**
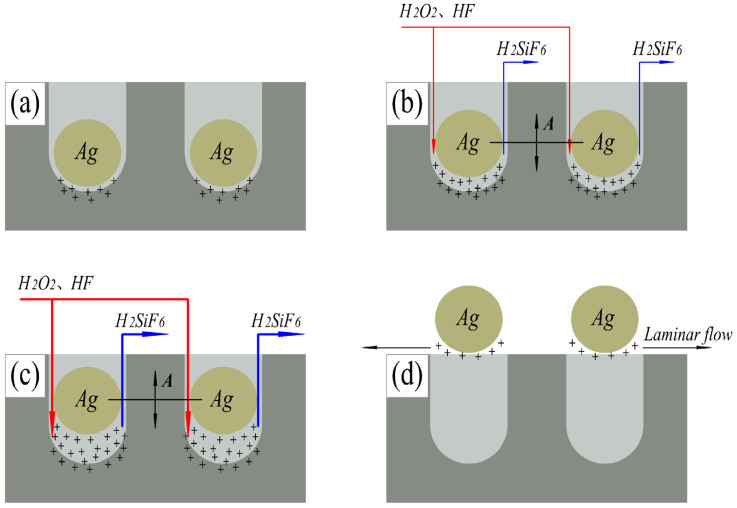
Etching mechanism model with no vibration (**a**), and with amplitudes of 2 μm (**b**), 4 μm (**c**), and 6 μm (**d**). The “+” in the figure represents a hole.

**Figure 4 materials-16-05483-f004:**
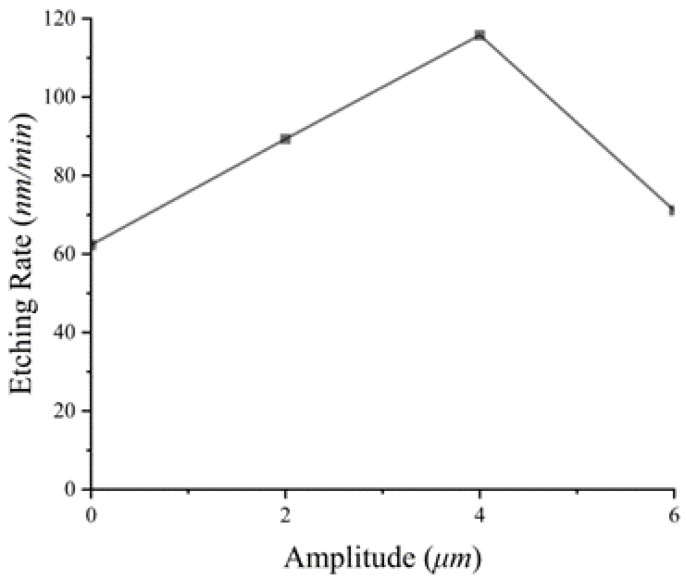
Etching rate of SiNWs at different amplitudes.

**Figure 5 materials-16-05483-f005:**
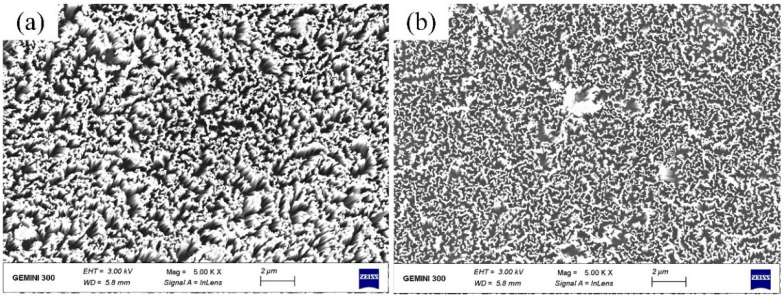
Plan view SEM image of SiNWs with no vibration (**a**), and with the vibration at the frequency of 20 Hz (**b**).

**Figure 6 materials-16-05483-f006:**
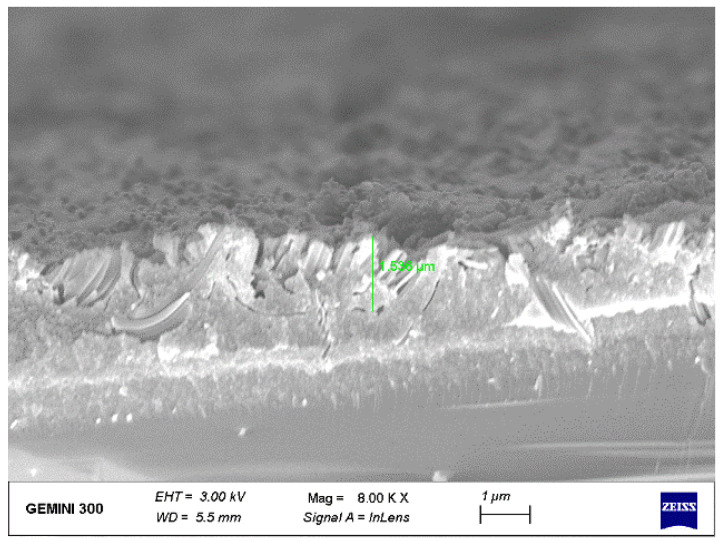
Plan view SEM image of SiNWs.

**Figure 7 materials-16-05483-f007:**
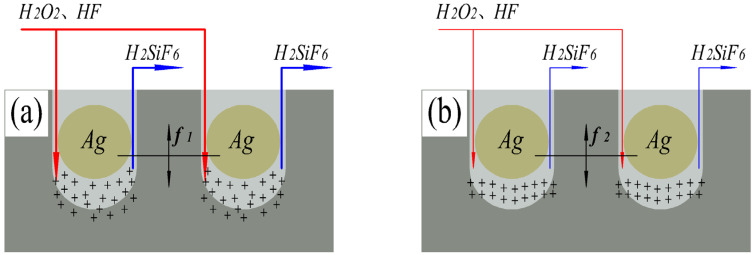
Etching mechanism model with frequencies of 20 Hz (**a**) and 40 Hz (**b**). The “+” in the figure represents a hole.

**Table 1 materials-16-05483-t001:** Parameter of signal generator.

Characteristic Parameter	FY2300-06M
Signal frequency range	0~6 MHz
Frequency minimum resolution	1 μHz
Output amplitude range (peak-to-peak)	5 mVpp~20 Vpp
Amplitude resolution	1 mV
Amplitude stability	±5%/5 h

**Table 2 materials-16-05483-t002:** Parameter of power amplifier.

Characteristic Parameter	FPA1016
Maximum output amplitude	100 Vpp
Input signal amplitude range	0~25 Vpp
Maximum output power	30 W

**Table 3 materials-16-05483-t003:** Experimental parameters.

HF (mol/L)	H_2_O_2_ (mol/L)	Amplitude (μm)	Frequency (Hz)	Etching Time (min)
4.8	0.2	/	/	30
4.8	0.2	2	20	30
4.8	0.2	4	20	30
4.8	0.2	6	20	30
4.8	0.2	2	40	30

## Data Availability

Data sharing is not applicable for this article.

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
