# Peer review of "Fabrication of Silicon Nanowires by Metal-Assisted Chemical Etching Combined with Micro-Vibration"

_materials, 2023, doi:10.3390/ma16155483_

Round 1

Reviewer 1 Report

In this work the authors studied the effects of the micro-vibration process combined with MaCE etching to realize Si based nanostructures.

The paper the article is suitable for publication to Materials after the following major changes - justifications.

Introduction:

The references about the MACE etching are lacking [15- 18]. The following articles are suggested to be included:

-Nanotechnology 24 (2013) 335302, doi:10.1088/0957-4484/24/33/335302

-Advanced Functional Materials, https://doi.org/10.1002/adfm.200900181

2. Micro-vibration Platform

Are the authors able to quantify the diffusion rate of the etchant?

3. Experimental

The studies were performed for p-doped silicon. Do the authors expect the same behaviour also for n-doped silicon and with a different substrate orientation?

A detail about the process scalability is suggested.

The nanostructures (wires) visible in the SEM images appear slightly tilted. Is this tilt due to vibration?

In figure 4 the authors report the etching rate of SiNWs at different amplitudes. The trend is obtained considering only 4 points. An increase in the measurements is strongly recommended, including the possible measurement error in the plot.

English can be improved

Reviewer 2 Report

Review Report: materials-2507309

The manuscript, #materials-2507309, entitled “Fabrication of Silicon Nanowires by Metal-assisted Chemical 2 Etching Combined with Micro-vibration” written by Weiye Huang, Junyi Wu, Wenxin Li, Guojin Chen, Changyong Chu, Chao Li, Yucheng Zhu, Hui Yang, Yan Chao. They fabricated silicon nanowires by metal-assisted chemical etching. By additional function with micro-vibrations, the etching rate was enhanced. This study has paid the attention to the scientific community and has the worth of publication in Materials (MDPI). This manuscript should be accepted in “Materials”, if authors clarify few queries and modified the manuscript with major corrections. The manuscript cannot be accepted in the present format.

1.     Can authors describe about the uniformity and controlling the aspect ratio SiNWs?

2.     Abstract is not written well, it must be re-written as per the investigation in the present research. First 3 sentences “Silicon nanowires (SiNWs) have exhibited promising application in the fields of microelectronics, optoelectronics, solar cells, chemical and biosensors. Metal-assisted chemical etching (MaCE) plays an important role in the preparation of SiNWs due to its characteristics of simplicity, versatility and cost-effectiveness as compared with other etching methods. However, the change of etching environment, including concentration of etchant and generation of etching production, will continuously influence the etching rate and the quality of SiNWs.” must be deleted in the revised manuscript. This is the part of introduction not for abstract. It will be helpful for making the objective of this research work.

3.     Highlights are not given to review the manuscript.

4.     Author should define the percentage error of Fig. 4.

5.     Many grammatical and syntax errors were observed, please revise the manuscript thoroughly and correct them before re-submission. Some sentences are not giving clear information, these must be re-write.

6.     Concluding remarks should be focused on the investigations not a general sentence or projected applications.

Minor revision.

Reviewer 3 Report

The authors combined the traditional MaCE process with micro-vibration conditions to prepare silicon nanowires. The length of SiNWs is improved by vibration amplitude and frequency. The mechanism of the micro-vibration combined etching process was proposed after analyzing the TEM images of SiNWs. It is highly useful for the fabrication of SiNWs. The manuscript can be published in Materials after minor revision:

1. The Introduction should add more references to highlight the advantages of micro-vibration conditions on SiNWs fabrication.

2. Do the authors have other methods to investigate the quality of SiNWs except SEM?

3. The name of Fig. 2 (a, b, c, and d) and Fig. 5 (a and b) should describe more detail.

Round 2

Reviewer 1 Report

The article has been enhanced and I consider it an appropriate publication for Materials journal.

Best regads,

Monica Bollani